# Stem Cell Theory of Cancer: Clinical Implications for Cellular Metabolism and Anti-Cancer Metabolomics

**DOI:** 10.3390/cancers16030624

**Published:** 2024-01-31

**Authors:** Shi-Ming Tu, Jim Z. Chen, Sunny R. Singh, Sanjay Maraboyina, Neriman Gokden, Ping-Ching Hsu, Timothy Langford

**Affiliations:** 1Division of Hematology and Oncology, University of Arkansas for Medical Sciences, Little Rock, AR 72205, USA; zchen3@uams.edu (J.Z.C.); srsingh@uams.edu (S.R.S.); 2Department of Radiation Oncology, University of Arkansas for Medical Sciences, Little Rock, AR 72205, USA; smaraboyina@uams.edu; 3Department of Pathology, University of Arkansas for Medical Sciences, Little Rock, AR 72205, USA; gokdenneriman@uams.edu; 4Department of Environmental & Occupational Health, University of Arkansas for Medical Sciences, Little Rock, AR 72205, USA; phsu@uams.edu; 5Department of Urology, University of Arkansas for Medical Sciences, Little Rock, AR 72205, USA; tlangford@uams.edu

**Keywords:** metabolism, cancer stem cells, glycolysis, OXPHOS, PGC-1, EMT, HIF1-alpha, glutamine, aspartame, ketogenic diet, GLP-1, metformin

## Abstract

**Simple Summary:**

We propose that having a proper perspective and narrative about the origin and nature of cancer metabolism affects cancer research and cancer care in integrated versus targeted therapy, multimodal versus precision medicine, and drug versus therapy development. Knowing and understanding whether cancer is a metabolic, genetic, or stem cell disease influences how we conduct informative cancer research and how we discover impactful cancer therapy.

**Abstract:**

Although Otto Warburg may be right about the role of glycolysis versus OXPHOS in cancer metabolism, it remains unclear whether an altered metabolism is causative or correlative and is the main driver or a mere passenger in the pathogenesis of cancer. Currently, most of our successful treatments are designed to eliminate non-cancer stem cells (non-CSCs) such as differentiated cancer cells. When the treatments also happen to control CSCs or the stem-ness niche, it is often unintended, unexpected, or undetected for lack of a pertinent theory about the origin of cancer that clarifies whether cancer is a metabolic, genetic, or stem cell disease. Perhaps cellular context matters. After all, metabolic activity may be different in different cell types and their respective microenvironments—whether it is in a normal progenitor stem cell vs. progeny differentiated cell and whether it is in a malignant CSC vs. non-CSC. In this perspective, we re-examine different types of cellular metabolism, e.g., glycolytic vs. mitochondrial, of glucose, glutamine, arginine, and fatty acids in CSCs and non-CSCs. We revisit the Warburg effect, an obesity epidemic, the aspartame story, and a ketogenic diet. We propose that a pertinent scientific theory about the origin of cancer and of cancer metabolism influences the direction of cancer research as well as the design of drug versus therapy development in cancer care.

## 1. Introduction

Nowadays, we worry that sugar is bad for our health and obesity may be linked to cancer. Metabolism is basic to our body functions and integral to our well-being. How to harness cellular metabolism for the sake of good health and target cancer metabolism for the purpose of drug development is a holy grail in clinical medicine and cancer care. 

There is no dispute that metabolism may be altered or aberrant in a cancer cell. But altered or aberrant in what and which cell is critical to the inquisitive mind and according to the scientific method. After all, an appropriate control makes all the difference in the interpretation of the results from an experiment. Metabolism may very well be altered or aberrant in a cancer cell compared to a normal cell but may not be so when comparing a cancer stem cell (CSC) with a normal stem cell (NSC)—especially when we consider that CSCs are related to and may be derived from NSCs, and they share similar metabolic needs and deeds.

In this perspective, we re-examine cancer metabolism according to a unified theory and a stem cell origin of cancer. We reassess different types of cellular and cancer metabolism, e.g., glycolysis and oxidative phosphorylation (OXYPHOS) and their metabolic substrates, e.g., glucose, glutamine, arginine, in various cell types, e.g., progenitor CSCs and progeny non-CSCs (Figure 1). We revisit the clinical implications of cellular and cancer metabolism in an obesity epidemic, the aspartame debate, and a ketogenic diet. We illustrate that a pertinent scientific theory about the origin of cancer and of cancer metabolism influences the direction of cancer research as well as the design of drug versus therapy development in cancer care. When we have a proper cancer theory at our disposal, we will make fewer cancer mistakes both in the present and for the future.

## 2. Brief History

In 1948, when Farber demonstrated that anti-folates provided anti-cancer effects in patients with acute leukemia [1], cancer metabolism was on the cusp of clinical reality and relevance (Table 1).

However, it was not until 1956 that Warburg first reported altered metabolism in cancer: increased aerobic glycolysis was maintained in conditions of high oxygen tension leading to enhanced lactate production [2]. He thought that cancer cells preferentially produce ATP by way of glycolysis rather than through oxidative phosphorylation (OXYPHOS), because of damage in the mitochondrial genome and a resultant dysfunctional Krebs cycle. 

In 2010, Seyfried revived the idea that cancer is a metabolic disease [3]. However, a prevailing dogma that cancer is a genetic disease still pervades and percolates in our minds thanks to the pioneering studies by Vogelstein and colleagues in the 1980s [4]. 

In many respects, there are still many lingering unknowns in either idea of cancer being a metabolic or genetic disease [5,6]. For instance, certain cancer cells preferentially utilize glycolysis despite having fully functional mitochondria [7]; non-cancer cells may harbor the same genetic defects that cancer cells have [8,9].

Perhaps Virchow was right after all. In 1863, he proposed a stem cell origin of cancer [10]. Subsequently, Stevens [11] and Pierce [12,13] demonstrated a stem cell origin in germ cell tumors. A stem cell theory of cancer embraces all aspects of cancer hallmarks, including the complexities of cancer metabolism and the vagaries of genetic instability [14,15]. Importantly, the idea that cancer is a stem cell disease rises above and beyond the metabolome and the genome—it envisions a unified theory of cancer that views cancer in a different cellular perspective and metabolism in the proper epigenomic context [14,15]. 

**Table 1 cancers-16-00624-t001:** Brief timeline of paradigm-shifting concepts of a metabolic, genetic, or stem-cell origin of cancer.

Name, Year	Origin/Nature of Cancer	Contribution	Reference
Virchow, 1863	Stem-ness	Embryonal cells	[10]
Rous, 1911	Genetic	Viral oncogenes	[5]
Farber, 1948	Metabolic	Antifolate therapy of acute leukemia	[1]
Warburg, 1956	Metabolic	Increased glycolysis and defective mitochondria	[2]
Stevens, 1964	Stem-ness	Origin of cancer stem cells	[11]
Knudson, 1971	Genetic	Tumor suppressor genes: 2-hit hypothesis	[6]
Vogelstein, 1988	Genetic	Multistep carcinogenesis	[4]
Pierce, 1994	Stem-ness	Maturation arrest of stem cells	[12,13]
Seyfried, 2010	Metabolic	Nutrition and cancer	[3]

## 3. Unified Theory of Cancer

### 3.1. Stem Cell Origin 

There is evidence suggesting that CSCs (a small subpopulation of cells within tumors capable of self-renewal, differentiation, and tumorigenicity when transplanted into an animal host) mimic or mirror NSCs, because the former may be related to, if not derived from, the latter [14,15]. Consequently, they share many stem-ness and stem-like properties, including metastatic potential, intra-tumoral heterogeneity, cancer dormancy, drug resistance, and cellular metabolism [16,17,18,19,20]. 

Because stem cells are primeval and tend to be quiescent cells, we postulate that they are more inclined to utilize a relatively primitive but cost-effective and safety-proficient machinery in the form of glycolysis rather than OXPHOS for their metabolic needs. 

Considering that embryonic and germ cells are prototype stem cells, we propose they are appropriate cellular models and should be preferred experimental controls to investigate and elucidate the clinical implications of cellular metabolism in CSCs and non-CSCs. 

Given that progenitor stem cells and progeny differentiated cells in both normal tissues and malignant tumors interact with one another and with their respective microenvironment, we recall that many epigenomic processes, including cellular metabolism, are dynamic rather than static, interconnected rather than isolated, and integrative rather than separated.

Therefore, we would like to elaborate that both CSCs and NSCs preferentially, if not predominantly, produce ATP via glycolysis rather than OXPHOS. We aim to illustrate that mitochondrial activity in these cells is likely to be inhibited rather than defective within malignant tumors.

### 3.2. Cellular Context

When it concerns the metabolome (and the epigenome, as well as the genome), cellular context is supreme and a prerogative. Metabolic requirements and activity may be different in different cell types and within their respective microenvironments—whether we are dealing with a normal progenitor stem cell (i.e., NSC) versus normal progeny differentiated cell, or with a malignant progenitor stem cell (i.e., CSC) versus malignant progeny differentiated cell (e.g., non-CSC). 

It remains unclear whether self-renewing malignant CSCs utilize aerobic glycolysis for most of their metabolic energy needs like their nonmalignant NSC counterparts. 

For example, long-term hematopoietic stem cells (LT-HSCs) use glycolysis instead of mitochondrial OXPHOS to meet their energy needs, because of their location in a hypoxic niche [21]. However, a whole tumor comprises a continuum of myriad cell types with variable dependency on aerobic glycolysis for their energy requirements and demands. In those real cancers in real patients whom we treat in the clinics rather than virtual tumor models which we study in the laboratories, tumoral heterogeneity indicates and implicates that we need to consider tumor metabolism in various tumor compartments, components, and the microenvironment. Put simply, tumors have a variable mixture of undifferentiated self-renewing CSCs that tend to use aerobic glycolysis and differentiated non-CSCs that tend to use OXPHOS. 

## 4. Cellular vs. Cancer Metabolism

### 4.1. Glycolysis 

Lactate dehydrogenase A (LDHA) is known to be a key regulator of glycolysis [22]. Hu et al. showed that an abundance and the activity of LDHA were tightly correlated with in vivo pyruvate conversion to lactate in malignant tumors [22]. Furthermore, conversion of pyruvate to alanine predominated in precancerous tissues prior to observable morphologic or histological changes and to MYC activity. 

Interestingly, many glycolytic genes in tumors were upregulated when MYC was turned on. The earliest metabolic change (preceding both tumor formation and regression) was increased alanine metabolism in heterogeneous regions that could represent the stem-ness tumor microenvironment (TME) or a stem-like onco-niche.

Similarly, San-Millan et al. [23] revealed that lactate acted like an oncometabolite that affected the transcription of certain oncogenes (*Myc*, *Ras*, and *Pi3kca*), transcription factors (HIF1A and E2F1), tumor suppressors (BRCA1, BRCA2), as well as cell cycle and proliferation genes (*Akt1*, *Atm*, *Ccnd1*, *Cdk4*, *Cdkn1a*, and *Cdk2b*). Dang et al. [24] reported that several oncogenes, including activated alleles of *Ras*, Akt, and *Pi3kca*, or loss of tumor suppressors (such as *p53* and *Vhl*) increased glycolysis.

It is plausible that these global metabolic changes correlate with glycolytic activity and perhaps with a stem-ness property that is quantifiable based on LDHA levels and imageable, such as with ^13^C-pyruvate magnetic resonance spectroscopy [22].

### 4.2. Mitochondrial Switch

Murine epiblast stem cells (EpiSC) resemble human embryonic stem cells (hESC) and CSCs, according to their transcriptional and translational profiles, i.e., their epigenetic state [25].

While ESCs are bivalent in their energy production, it is of interest that EpiSCs and hESCs are almost exclusively glycolytic [25]. Furthermore, EpiSCs have more mature mitochondria and mtDNA copy numbers, even though their mitochondria are paradoxically less active due to compromised COX activity. Etymologically, this would be advantageous because less aerobic respiration should minimize potential harm generated by reactive oxygen species (ROS) from mitochondrial activities and protect all vital germ lines arising from hESCs and EpiSCs.

Like EpiSCs and hESCs, B16 metastatic melanoma is also devoid of functional mitochondrial electron transport and completely dependent on glycolytic ATP for energy production. However, in contradiction to Warburg’s hypothesis, there was a delay in tumor growth and no lung metastasis formation in this malignant tumor model [26]. 

Intriguingly, peroxisome proliferator-activated receptor gamma coactivator-1 (PGC-1) may be central to solving this paradox of stem-ness and metabolism. After all, PGC-1 regulates mitochondrial activity. Coordinated regulation of PGC-1 enables early embryonic cells to develop sufficient mitochondria as a reservoir for the increased energy demands of future differentiation, while at the same time maintaining anaerobic metabolism crucial for self-renewal and pluripotency.

### 4.3. HIF1α—Master Regulator

Hypoxia-inducible factor (HIF) 1α is a key regulator of the pluripotent state and in the metabolic and functional transition from ESCs to EpiSCs. After all, embryonic development occurs in a hypoxic environment [27,28]. HIF1α overexpression induces morphogenetic changes reflective of the transition from ESC to EpiSC. It also induces active suppression of mitochondrial oxidative respiration. Metabolic changes during ESC-to-EpiSC transition induced by HIF1α act through *Activin/Nodal* signaling. 

Hence, HIF1α stabilizes *Activin/Nodal* and inhibits *c-Myc*, thereby negatively regulating PGC-1β and actively repressing mitochondrial activity. Activation of *Activin/Nodal* signaling is required to maintain pluripotency (as in EpiSCs) and prevent spontaneous differentiation.

Therefore, HIF1α is a master regulator and metabolic switch [29]: it plays an important role not only in anaerobic metabolism by activating key glycolytic enzymes, but also in OXOPHOS by repressing mitochondrial activity through inhibition of PGC-1β. Moreover, HIF1α also acts through *Activin/Nodal* signaling to maintain pluripotency and inhibit differentiation. 

## 5. Metabolic Substrates

### 5.1. Glucose

Warburg thought that the main source of carbon for lactate was glucose. He hypothesized that cancer cells preferentially produced ATP via glycolysis rather than the Krebs cycle, because of damage in the mitochondrial genome and a dysfunctional Krebs cycle.

When the Krebs cycle is being forced to operate in a tumor cell, such as by using dichloroacetate (DCA), which inhibits pyruvate dehydrogenase and increases flux of pyruvate into the mitochondria and shifting metabolism from glycolysis to oxidation, it can promote tumor cell death [30]. Similarly, 2-deoxyglucose (2DG), a glucose analogue that can be phosphorylated to 2-deoxy-6-phosphate but cannot be metabolized further, is proapoptotic [31]. 

However, neither DCA nor 2DG appeared to be effective for the treatment of patients with cancer in two phase I clinical trials [32,33]. Unfortunately, what works in the laboratory often enough does not work in the clinic, especially when we operate under a false or faulty hypothesis about cancer’s origins and metabolism and do not adopt or adhere to the proper scientific method [34]. 

Although Warburg might be right about altered glycolysis versus OXPHOS in the pathogenesis of cancer, he could not distinguish whether the putatively defective metabolism was causative or correlative, the main driver or a mere passenger in the whole story and big picture of cancer. Perhaps cellular context matters after all. We wonder whether aberrant metabolism in a progenitor CSC versus a progeny non-CSC is at play and ponder whether the same mechanisms of action in different cell types may be pivotal in our investigation and elucidation of cancer metabolomics in cancer care.

### 5.2. Glutamine

It turns out that glutamine (rather than glucose) is a major source of carbon in the production of lactate within certain cell types (such as the ova) and many tumor types [35,36]. 

During anaplerosis (metabolic pathways that replenish intermediates for the Krebs cycle), mitochondrial glutamate dehydrogenase 1 (GLUD1) plays a key role in the conversion of glutamate to alpha-ketoglutarate (α-KG) [37]. α-KG is generated for the Krebs cycle and is used for OXPHOS. GLUD1 is overexpressed in various cancer cells, promoting epithelial–mesenchymal transition (EMT) and drug resistance. Furthermore, MYC drives glutaminolysis by upregulating GLUD1. 

Lu et al. [38] found that glutamine-dependent mTORC1 signaling pathway superseded lineage-specifying cytokine induction in the ectoderm. In contrast, glutamine served as the preferred precursor of α-KG without a direct role in cell fate signaling in the mesoderm and endoderm.

Similarly, Vardhana et al. [39] demonstrated that reduced dependence on glutamine anaplerosis is an inherent feature of self-renewing pluripotent stem cells. However, transient glutamine withdrawal led to selective elimination of non-pluripotent cells. 

Kim et al. [40] showed that hair follicle stem cells (HFSC) were metabolically flexible, and the progenitor state required glutaminolysis. Progenitor fate reversibility required mTORC2-driven attenuation of glutamine metabolism. Hence, mTORC2 deletion impaired niche regeneration by progenitor cells, triggering HFSC exhaustion.

We propose that if CSCs mimic NSCs, then lessons from one may inform about the other. If glutamine plays a significant role in maintaining the stem-ness of certain cancer cells in a CSC hierarchy, then it is understandable that L-asparaginase is therapeutic for cancer care in part because it inhibits glutamine metabolism and/or depletes glutamine, thereby enabling the control if not elimination of those CSCs [41]. Similarly, since cancer-associated fibroblasts, adipocytes, and senescent cells in the TME could indirectly influence CSC fate by modulating glutamine availability, it is conceivable that treating the CSC niche would be just as important as treating the CSC itself [42]. 

### 5.3. Arginine

It makes sense that both CSCs and NSCs have a similar, if not the same, capacity to recycle arginine in polyamine metabolism to serve their stem-ness needs and purposes. Importantly, this capability may be empowered through certain stem-like genes, such as RBM39 [43]. 

Polyamines (e.g., spermidine, spermine) are ubiquitous small basic molecules involved in various vital cellular functions, including ion channels, cell–cell interactions, the cytoskeleton, and signaling via phosphorylation. Zhao et al. [44] showed that high polyamine levels promote ESC self-renewal. Furthermore, regulators of the polyamine pathway (e.g., *Amd1* and *Odc1*) can partially substitute for *Myc* during cellular reprogramming. 

Mossmann et al. [43] reported that arginine levels are elevated in murine and patient hepatocellular carcinoma, despite reduced expression of arginine synthesis genes. Tumor cells accumulated high levels of arginine due to increased uptake and reduced arginine-to-polyamine conversion. Importantly, high levels of arginine promoted tumor formation via metabolic reprogramming, including changes in glucose, amino acid, nucleotide, and fatty acid metabolism. They demonstrated that RBM39-mediated upregulation of asparagine synthesis led to enhanced arginine uptake, creating a positive feedback loop to sustain high arginine levels and oncogenic metabolism. 

Interestingly, Rana et al. [45] connected polyamine metabolism with mesenchymal (i.e., EMT) gene signature in the most aggressive, invasive, and multitherapy-resistant subtype of brain cancer, namely glioblastoma multiforme, with CSC features. Khan et al. [46] showed the therapeutic potential of targeting both polyamine synthesis and uptake in an incurable brain cancer, diffuse intrinsic pontine glioma. Importantly, Chen et al. [47] demonstrated that inhibition of RBM39 using indisulam led to inhibition of KRAS4A tumorigenicity through CSCs.

### 5.4. S-Adenosyl-L-methionine (SAM)

In early development, consumption of SAM by nicotinamide N-methyltransferase (NNMT) makes it unavailable for histone methylation, resulting in an altered epigenetic landscape within hESCs [48]. In some cancers, NNMT mediates EMT [49] and induces resistance to apoptosis via the mitochondrial pathway [50]. 

In addition, the activity of NNMT is tightly linked to the maintenance of nicotinamide adenine dinucleotide (NAD^+^), which modulates multiple enzymatic reactions that affect redox metabolism, mitochondrial functions, stem-ness properties, autophagic processes, cellular stress, ion homeostasis, and the circadian rhythm [51,52]. This is reminiscent of a complex metabolic network in need of a unified conceptual framework, as premised in this perspective [14,15]. Importantly, NNMT may be a potential biomarker and therapeutic target for cancer diagnosis and treatment [53,54,55].

## 6. Clinical Implications

### 6.1. Aspartame Saga

In 1996, Olney et al. raised the specter of a link between aspartame and brain tumors in humans [56]. This led to intensive studies that showed high consumption of aspartame in rats caused malignant tumors in multiple organs, including the kidneys, breasts, and the nervous system [57,58]. Of concern were the findings that the effects of aspartame were noticeable at low doses and in utero [58]. More recent studies confirmed and reinforced previous results [59].

Metabolically, it is difficult to attribute any putative carcinogenic effects of aspartame directly to its breakdown products, namely, aspartic acid, phenylalanine, and methanol. After all, aspartic acid and phenylalanine are mere amino acids that seem relatively innocuous and ubiquitous. Although the body may convert methanol to formaldehyde, which is a known carcinogen and neurotoxin, the same process occurs when we consume fruits, fruit juices, some vegetables, and fermented beverages. For example, tomato juice may lead to the formation of about 5–6 times higher levels of methanol in the body than aspartame-sweetened beverages do [60].

Interestingly, Gezginci-Oktayoglu et al. [61] showed that long-term aspartame (but not high glucose) exposure increases the CSC population and tumor cell aggressiveness through p21, NICD, and GLI-1. The EMT marker N-cadherin increased only in the aspartame, but not high glucose, group. High levels of aspartame but not glucose exposure increased invasion and migration of PANC-1 cells. Moreover, while aspartame had no direct tumorigenic effect, it could potentially advance an existing tumor. Similarly, Pontel et al. [62] demonstrated that formaldehyde is a genotoxin and metabolic carcinogen to hematopoietic stem cells.

Perhaps chronic exposure to aspartame particularly at high levels makes all the difference whether malignancy occurs or not in vulnerable individuals. According to the stem cell theory of cancer, the type of cell and its microenvironment being affected also matter. Perhaps there are other factors that afflict health-agnostic people who consume more aspartame over their lifetime than those health-conscious individuals who drink plenty of tomato juice and eat more than their share of fruits and vegetables. Perhaps there are other ingredients in a healthy and balanced diet that protect us from the effects of stem cell genotoxins and metabolic carcinogens. Perhaps what happens to rats and mice may not happen to humans, because we are not rodents.

### 6.2. Lipid Phobia

Obesity is a pressing public health issue in modern society. It is evident that lifestyle changes such as diet, exercise, and behavior can affect both the occurrence and management of obesity. 

According to Pati et al. [63], about 4–8% of all cancers can be attributed to obesity. Obesity is associated with multiple common malignancies, including breast, colorectal, kidney, esophageal, gallbladder, uterine, pancreas, and liver cancers. Obesity is associated with not only an increased risk of cancer, but also risk of cancer recurrence and mortality among cancer survivors. 

A direct link between obesity and malignancy remains elusive. Assuming that obesity may be linked to metabolism, then appropriate management of metabolism may provide tangible benefits in the care of both obesity and malignancy. Alvina et al. [64] showed that CSCs, non-CSCs, and NSCs utilize similar metabolic mechanisms and pathways. Their results suggested that development of a specific metabolic drug that selectively targets CSCs and non-CSCs but spares NSCs in cancer care might be quixotic. On the other hand, a specific metabolic drug that addresses both cancer and non-cancer cells, as well as CSCs and non-CSCs, would be pragmatic (especially if it is also inexpensive and nontoxic), because we manage to keep both cancer at bay and obesity in check. 

Although tampering with aberrant or altered lipid metabolism is conceptually tempting, tempering aberrant or altered CSCs (as well as non-CSCs) may be paradigm-shifting. After all, drug development based on precision medicine tends to focus on lipid metabolism in cancer cells rather than on various types of cancer cells. In contrast, therapy development based on a stem cell theory of cancer focuses on an integrated approach in which one treats both CSCs (and non-CSCs) with their unique (or lack of) stem-ness properties, including cancer metabolomics, along with their obligate onco-niches (such as by modulating or reducing NFkB signaling) [65]. 

### 6.3. Ketogenic Diet

Ketogenesis is a metabolic process in which the body produces ketone bodies as an alternative fuel source to glucose. This process occurs when the body is in a state of low carbohydrate availability, such as during fasting or a ketogenic diet. 

In a ketogenic diet, one receives more calories from proteins and fats and less from carbohydrates. One deliberately avoids those carbohydrates that are easy to digest, such as sugar, sodas, pastries, and white bread.

Studies have suggested that ketogenesis and a ketogenic diet may provide anti-cancer effects by limiting the availability of glucose and other nutrients that feed cancer cells and promote their growth and proliferation [66].

Chi et al. [67] found a relationship between increased concentrations of ketosis-related compounds and prostate specific antigen (PSA) double time, indicating that cancer growth was reduced in a ketosis-intensified diet (CAPS2 diet trial).

Because current evidence supports a plant-based diet as part of a healthy lifestyle associated with reduced cancer risk [68], how do we reconcile a plant-based with ketogenic diet when it concerns cancer prevention and/or management? Perhaps it will take a randomized trial comparing a plant-based vs. fat-based diet to prove which is better for patients with prostate and other cancers.

It is of interest that several natural compounds produced during ketosis, such as beta-hydroxybutyrate, a short-chain fatty acid with documented antioxidant and anti-inflammatory effects, may affect the microbiome in a favorable manner and provide salubrious as well as anti-cancer effects.

For example, Cheng et al. [69] reported that both fasting and a high-fat diet improve the function of intestinal stem cells. They published that any kind of diet in which carbohydrate intake is limited stimulates ketogenesis and promotes the growth of intestinal stem cells.

What remains unclear is that if fasting or a ketogenic diet enables healing and proliferation of intestinal NSCs, would it paradoxically empower CSCs, too? Perhaps, ketogenesis and fasting are good for prevention of cancer, but not so good when a patient already has active, if not fulminant, cancer.

It seems common sense that losing weight and fasting may be harmless for some overweight patients with cancer and subjects without cancer. However, when cancer patients are already cachexic and anorexic, losing more weight does not make sense at all and is likely to be harmful.

Therefore, an all-important question needs to be addressed: What is the proper clinical setting in which a ketogenic diet may have utility for a patient with cancer, assuming that it does favorably impact carcinogenesis?

Ironically, when the question to be answered and a hypothesis to be tested are necessarily relevant and more specific (e.g., does it affect non-CSCs vs. CSCs, is it preventive vs. palliative), negative results from a research study and unfavorable outcomes from a clinical trial are still more than informative and impactful, because we learn that what is popular may not be proper and we know when not to consume or pursue a ketogenic diet. 

According to a stem cell theory of cancer, we predict that when the CSC is deprived of glucose, it will resort to another source or other resources, such as glutamate and fatty acids instead. Importantly, if ketogenesis and a ketogenic diet empower CSCs like they do for NSCs, then the overall clinical impact could be potentially harmful in the wrong clinical settings for certain cancer patients.

### 6.4. Weight Loss Shots or Pills

Another way to manipulate metabolism and to lose weight is through glucagon-like peptide-1 (GLP-1) [70]. As an example, semaglutide is a GLP-1 receptor agonist (GLP-1RA) that directly activates POMC/CART neurons and indirectly inhibits NPY/AgRP neurons that collectively result in reduced food intake [71,72]. One may lose about 15% of one’s weight on semaglutide (vs. 2.4% on placebo, STEP-1 trial) [73].

Similarly, tirzepatide is a dual GLP-1R and glucose-dependent insulinotropic peptide (GIP) agonist that improves beta-cell function and insulin sensitivity in patients with type 2 diabetes mellitus. One loses 20% or more of one’s weight (vs. 3% on placebo, SURMOUNT-1 trial) on tirzepatide [74].

Interestingly, metformin also has a direct and AMPK-dependent effect on GLP-1-secreting intestinal L cells and increases postprandial GLP-1 secretion, which contributes to its glucose-lowering and weight-reducing effects. Patients taking metformin may lose about 2% of their weight (vs. placebo 0.2%) [75].

Emerging evidence indicates that GLP-1 RAs elicit multifactorial effects—they exert direct and indirect salutary effects on the neurological system (Alzheimer’s, Parkinson’s, stroke, chronic pain), cardiovascular system (atherosclerosis, hypertension), and endocrine metabolic system (polycystic ovarian syndrome, obesity, non-alcoholic fatty liver disease). Unexpectedly (to many people), they may also affect cancer through the developmental system by way of stem-ness/stem-like pathways [76].

Fortunately, laboratory and clinical studies so far suggest that GLP-1 RAs do not appear to increase the risk of breast, pancreatic, or thyroid tumor formation. On the contrary, they may exert anti-tumor effects in the management of prostate, breast, pancreatic, and cervical cancers [76].

Interestingly, GLP-1/Notch signaling plays a role in germ-line stem cell maintenance [77,78]. Sforza et al. [79] showed that CD34+ hematopoietic stem progenitor cells (HSPCs) express GLP-1R at the transcriptional and protein levels. They speculated that GLP-1 RAs provide cardiovascular protective effects by improving CD34+ HSPC function. Similarly, Sanz et al. [80] demonstrated GLP-1R expression in human BM-derived mesenchymal stem cells (MSCs), whereas Lee et al. [81] showed that GLP-1 stimulated osteoblast differentiation but inhibited adipocyte differentiation in adipose-derived stem cells (ADSCs).

Again, we would like to pose a curious scientific question (hypothetical) with profound clinical implications (prophetical): Would GLP-1 RAs enable healing and proliferation of normal HSPCs, MSCs, and ADSCs, but paradoxically empower CSCs, too? Perhaps, GLP-1R is good for those people without cancer by protecting their NSCs and preventing cancer, but not so good for those patients with advanced, fulminant cancer laden with active CSCs.

## 7. More Therapeutic Implications

A stem cell theory of cancer stipulates that we need to treat the different compartments, components, and the microenvironment of cancer [82]. Currently, most of our successful treatments that target cancer metabolism are designed to eliminate non-CSCs, such as differentiated cancer cells [83,84]. When the treatments also happen to control CSCs or the stem-ness niche, it is often unintended, unexpected, or undetected for lack of a pertinent theory about the origin of cancer that clarifies whether cancer is a genetic, metabolic, or stem cell disease.

If cancer has a stem cell origin and is a stem-ness disease, then it is crucial that we have the means or devise ways to identify, monitor, and measure its presence and activity. For example, EMT alludes to stem-like properties; an inflammatory milieu may disturb or disrupt the stem-ness niche. Consequently, treatments that ameliorate EMT and attenuate inflammation may improve standard treatments that control non-CSCs, as in a preventive setting to avoid progression of any minimal cancer-initiating cells or in a maintenance regimen to delay recurrence of any minimal residual cancer cells.

### 7.1. An Exemplary Anti-CSC Drug

#### Metformin

In many respects, metformin is a forgotten, if not forsaken, anti-CSC agent. It modulates a network of stem-ness pathways and myriad stem-like processes. It manipulates both intracellular and extracellular modules rather than just singular or insular molecular silos. 

In 2005, Evans et al. published a landmark study that indicated reduction in the risk of subsequent cancer diagnosis in patients with type 2 diabetes mellitus (DM) who received metformin [85].

Subsequently, Richards et al. [86] showed that patients who received androgen deprivation therapy for their advanced castration-sensitive prostate cancer and metformin for their DM had a significant decreased risk of death (18%) and decreased risk of cancer-related death (30%) compared with those who did not have DM. Metformin use was associated with decreased mortality, even when compared with diet-controlled patients [87].

The reputed anti-cancer effects of metformin appear to be universal, i.e., affect many cancer types [88], and likely to be more pronounced under the right conditions, e.g., in an adjuvant setting [89]. If metformin does provide anti-cancer effects, it more likely than not does so through anti-metabolic mechanisms. But is anti-metabolism the only way metformin tackles cancer, or are its anti-cancer effects operative through multiple ways and pathways (e.g., anti-metabolic [90,91], anti-CSC [92,93], anti-EMT [94,95], pro-senescent [96], pro-differentiating [97], and/or multi-pronged [98,99,100,101,102])? Again, the answer to this profound question depends on our basic understanding of cancer—its origins and nature. Is cancer a metabolic, genetic, or stem-ness disease? Do we treat cancer by correcting its metabolic aberrations, overcoming its genetic defects, and/or managing its stem-ness problems? 

Therefore, one may envision metformin as an anti-epigenomic or an anti-metabolomic drug targeting CSCs. In many respects, metformin is a relatively “dirty” drug (like chemotherapy) rather than a “precision” medicine, because it targets myriad pathways (involving EMT, senescence/autophagy, etc.) in CSCs. This may be precisely or unwittingly (depending on our perspective of cancer) its therapeutic potential and healing power—it targets a whole system of cellular networks rather than just an isolated molecular target or signaling pathway in a pertinent tumor entity and phenotype. 

### 7.2. A Neglected Anti-CSC Drug

#### MAOA Inhibitors

Monoamine oxidase A (MAOA) is a mitochondria-bound enzyme that catalyzes the degradation of monoamine neurotransmitters (norepinephrine, epinephrine, serotonin, and dopamine) and dietary amines by oxidative deamination, which produces a by-product, hydrogen peroxide, a major source of ROS. As mentioned earlier, ROS can predispose cancer cells to DNA damage and cause tumor initiation and progression.

Interestingly, MAOA also induces EMT through activation of VEGF and its coreceptor neurophilin-1, and stabilizes HIF1α, which mediates hypoxia through elevation of ROS. MAOA-dependent activation of neurophilin-1 promoted AKT/FOXO1/TWIST1 signaling [103].

Inhibition of MAOA and MAOB using monoamine oxidase inhibitors is used to treat depression, erectile dysfunction, and anxiety.

Intriguingly, does this imply that treating depression, anxiety, and stress just like treating metabolism is not only good for our health but may also be good for our anti-cancer strategies? Indeed, would patients who receive MAO inhibitors for their depression and erectile dysfunction also fare better with their prostate cancer [103,104]? Perhaps one should not be surprised and would be able to predict that curcumin (present in turmeric, well known for its anti-inflammatory effects, but unknown as a MAO inhibitor) could elicit anti-cancer activity [105,106].

## 8. Drug vs. Therapy Development 

How and what do we learn from history so that we do not make the same mistakes and will have a greater chance of ensuring success rather than enduring failure in drug versus therapy development for cancer care [34]? We propose that having the right cancer theory about the origin and nature of cancer is imperative. Knowing and understanding whether cancer is a genetic, metabolic, or stem cell disease influences how we perform meaningful cancer research and how we discover impactful cancer therapy.

Hence, treatments that aim to control CSCs and target EMT are expected to be less effective if not ineffective for the control of non-CSCs that may be at play in fulminant cancers when the bulk of non-CSCs are rapidly proliferating and actively threatening. Consequently, one would predict that use of metformin in the wrong clinical settings based on the wrong scientific theories about the origin and nature of cancer (e.g., whether cancer is a genetic, metabolic, or stem-cell disease) could be misdirected [107], because after all, when the basic premise of a hypothesis is wrong, the result is likely to be misleading even though the experiments designed to test it may be infallible [34,108].

### 8.1. Tirapazamine 

For example, tirapazamine is a hypoxia-activated prodrug that kills hypoxic cells by inducing chromosome defects and DNA breaks. Therefore, it is likely to be more effective for the treatment of CSCs and less effective for the treatment of non-CSCs [109]. The story of tirapazamine explains a common finding from preclinical studies and early clinical trials in which preliminary results were more than encouraging and promising, but those from definitive randomized phase 3 trials (SWOG S0003, HeadSTART) using the same drug often showed marginal clinical benefits [110,111]. When we do not know about or understand the difference between CSCs and non-CSCs, we may design an anti-CSC treatment for non-CSC tumors (and vice versa) and either detect no clinical benefits for obvious reasons or incremental clinical efficacy for the wrong reasons. 

### 8.2. Belzutifan

Similarly, belzutifan may be effective by inhibiting HIF1α and controlling CSCs in VHL-associated and clear-cell RCC (LITESPARK-005) [112,113]. However, any clinical benefit from the use of belzutifan, a presumed anti-CSC drug, may be limited for the treatment of many if not most cancers that also contain a predominance of non-CSCs. We propose that one can enhance the utility of an anti-CSC drug, such as belzutifan, by combining it with another anti-CSC and/or with an anti-non-CSC treatment. 

In many respects, this is the essence of integrated versus precision medicine in a stem cell versus genetic or metabolic theory of cancer. We propose that multimodal is more likely than targeted therapy to provide a superior clinical outcome and improved cancer care by treating both CSCs and non-CSCs in most patients with heterogeneous cancers in the clinics rather than relying on preclinical results based on selected tumor models with relatively homogeneous cancer cells in the laboratories. 

## 9. Conclusions

According to the scientific method, we design experiments to test hypotheses and refrain from using the results of the experiments to generate hypotheses [34,108].

Conventional clinical observations and exceptional clinical experiences empower us to formulate a pertinent scientific theory about the role of glycolysis, mitochondrial switch, hypoxia, and EMT in cancer metabolism.

If we hypothesize that cancer is a metabolic disease and glucose is central to cancer metabolism, then we give DCA and 2DG and we take aspartame and ketogenic diet. We have learned and will discover that what is effective in treating cancer cells in the laboratories may be futile and could even be harmful to patients in the clinics [34]. 

If we hypothesize that cancer is a stem-cell rather than genetic or metabolic disease, then we need to consider genetic content and metabolic activity in the proper cellular context. We need to account for glycolysis and OXOPHOS, as well as glucose, glutamine, arginine, and other metabolic substrates and cancer metabolites in the right metabolic perspective and narrative (Table 2). We need to treat both CSCs, non-CSCs, and the onco-niche.

A proper perspective and narrative about the origin and nature of cancer affect cancer research and cancer care in drug versus therapy development, integrated versus targeted therapy, and multimodal versus precision medicine, and influence how we design and utilize novel anti-HIF1αs and GLP-1RAs or repurpose the traditional metformin and recycle the conventional MAO inhibitors in cancer care. 

When we formulate the right scientific theory and adopt or adhere to the proper scientific method in scientific research, it will be less likely, if not unlikely, that we will continue to make the same old mistakes or commit more new mistakes.

## Figures and Tables

**Figure 1 cancers-16-00624-f001:**
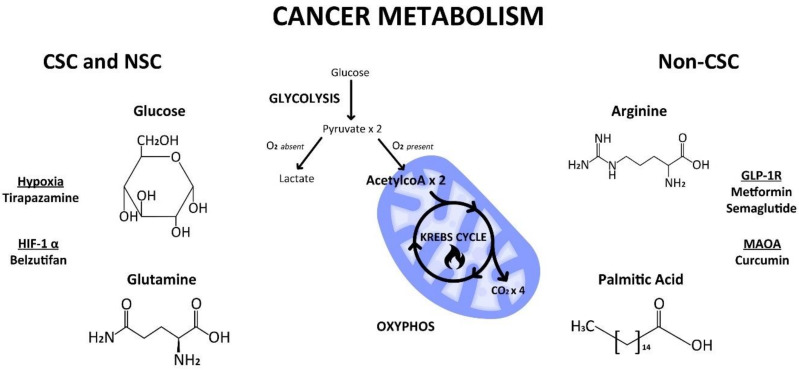
Cellular metabolism of glucose, glutamine, arginine, and palmitic acid by way of glycolysis in the cytoplasm or oxidative phosphorylation (Krebs cycle) in the mitochondria in progenitor cancer stem cells (CSC) and/or progeny non-CSC. Created by Benjamin Tu (www.bentubox.com), for this article on 2 December 2023.

**Table 2 cancers-16-00624-t002:** A purview of cancer metabolism: treatments and targets, according to a stem cell origin of cancer as discussed in this perspective. For additional examples, please refer to [82,83,84].

Metabolism	Treatments	Mechanisms/Targets	References
Glucose	Dichloroacetate	Pyruvate dehydrogenase	[32]
	2-deoxyglucose	Hexokinase 2	[33]
Glutamine	L-asparaginase	Beta-catenin	[41]
Arginine	Indisulam	RBM39	[47]
S-adenosyl-L-methionine	GYZ-319	NAD+/NNMT	[48,49,50,51,52,53,54,55]
Lipids	Ketogenic diet	Ketone bodies	[67]
	Semaglutide	GLP-1RA	[73]
	Tirzepatide	GIP and GLP-1RA	[74]
Glycolysis	Metformin	Nrf	[91]
		EMT	[94,95,96]
		miRNA let-7	[97]
		HER2	[98,99]
		Cyclin D1	[100,101]
		AMPK, mTOR	[102]
Monoamines	MAOAi	AKT/FOXO1/TWIST1	[103,104]
	Curcumin	COX-2, NFkB	[105,106]
Hypoxia	Tirapazamine	Hypoxia	[109,110,111]
	Belzutifan	HIF1-alpha	[112,113]

## Data Availability

The data presented in this article are available in the references provided.

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
