# Peer review of "Stem Cell Theory of Cancer: Clinical Implications for Cellular Metabolism and Anti-Cancer Metabolomics"

_cancers, 2024, doi:10.3390/cancers16030624_

Round 1
Reviewer 1 Report (Previous Reviewer 3)
Comments and Suggestions for Authors
Since the article has been resubmitted, I would like to see responses to reviewer concerns made during the previous round of review. Despite the absence of this data, it is clear that the authors significantly revised the manuscript; it became more structured and logically constructed. The authors took my comments into account in full.
Small notes:
1. What does "5.0 Mitochondrial Switch" mean on line 156?
2. I’m not sure that an epigraph is appropriate.
Author Response
We thank the reviewers for your insightful comments and suggestions. We have provided our responses below in bold with changes highlighted in red underneath the reviewer comments. We have incorporated all changes recommended by the reviewers with the changes highlighted in red in the manuscript.
Reviewer 1
Since the article has been resubmitted, I would like to see responses to reviewer concerns made during the previous round of review. Despite the absence of this data, it is clear that the authors significantly revised the manuscript; it became more structured and logically constructed. The authors took my comments into account in full.
We provide below the responses to reviewers 2 and 3 during the previous round of review, as requested.
Reviewer 2
The manuscript presents a perspective on the stem cell theory of cancer and its implications for understanding and targeting cancer metabolism. It emphasizes the impact of critical metabolic pathways on cancer progression, such as glycolysis, mitochondrial activity, and hypoxia. Additionally, it discusses the potential therapeutic implications of targeting cancer metabolism. Overall, the manuscript offers valuable insights into the complex interplay between cancer and cellular metabolism, highlighting the potential for novel therapeutic approaches based on the stem cell theory of cancer. More explicit guidance on practical applications and potential future research directions would enhance the manuscript’s value to the broader scientific community.
We thank the reviewer for your insightful and expert comments!
We have restructured the paper and dedicated Section 8.0 Drug and Therapy Development to your important specification of “explicit guidance on practical applications and potential future research directions.” This section provides a general guidance to this directive as referenced in [34] and includes some specific examples as related to cancer metabolism.
- Tu SM, Singh S, Arnaoutakis K, Malapati S, Bhatti SA, Joon AY, Atiq OT, Posters LL. Stem cell theory of cancer: Implications for translational research from bedside to bench. Cancers 2022; 14:3345.
Reviewer 3
There is currently no unified theory of the origin of cancer that clarifies whether cancer is a metabolic, genetic, or stem cell disease. The authors consider different types of cellular metabolism, such as glycolytic and mitochondrial, with glucose, glutamine, arginine and fatty acids in RSCs and non-ROCs. The authors suggested that the relevant scientific theory of the origin of cancer and cancer metabolism influences the direction of cancer research, as well as the development of drugs and cancer treatments.
- The article does not have an abstract. Add please.
Apologize that the abstract may have been misplaced.
Abstract
Although Otto Warburg may be right about the role of glycolysis versus OXPHOS in cancer metabolism, it remains unclear whether an altered metabolism is causative or correlative and is the main driver or a mere passenger in the pathogenesis of cancer.
Currently, most of our successful treatments are designed to eliminate non-cancer stem cells (non-CSC) such as differentiated cancer cells. When the treatments also happen to control CSC or the stem-ness niche, it is often unintended, unexpected, or undetected for lack of a pertinent theory about the origin of cancer that clarifies whether cancer is a metabolic, genetic, or stem cell disease.
Perhaps cellular context matters. After all, metabolic activity may be different in different cell types and their respective microenvironments – whether it is in a normal progenitor stem cell vs progeny differentiated cell and whether it is in a malignant CSC vs non-CSC.
In this perspective, we re-examine different types of cellular metabolism, e.g., glycolytic vs mitochondrial, of glucose, glutamine, arginine, and fatty acids in CSCs and non-CSCs. We revisit the Warburg effect, an obesity epidemic, the aspartame story, and a ketogenic diet. We propose that a pertinent scientific theory about the origin of cancer and of cancer metabolism influences the direction of cancer research as well as the design of drug versus therapy development in cancer care.
- There are not enough pivot tables and/or diagrams. So, I would at least add others to the historical excursion.
Added Table 1 with an expanded excursion to Brief History, as recommended.
“In 1948, when Farber demonstrated that anti-folates provided anti-cancer effects in patients with acute leukemia [1], cancer metabolism was on the cusp of clinical reality and relevance.
However, it was not until 1956 when Warburg first reported altered metabolism in cancer:”
Perhaps Virchow was right after all. In 1863, he proposed a stem cell origin of cancer [10]. Subsequently, Stevens [11] and Pierce [12,13] demonstrated a stem cell origin in germ cell tumors.
Table 1. Brief timeline of paradigm-shifting concepts on a metabolic, genetic, or stem-cell origin of cancer.
|
Name, year |
Origin/nature of cancer |
Contribution |
Reference |
|
|
|
|
|
|
Virchow, 1863 |
Stem-ness |
Embryonal cells |
10 |
|
Rous, 1911 |
Genetic |
Viral oncogenes |
5 |
|
Farber, 1948 |
Metabolic |
Antifolate therapy of acute leukemia |
1 |
|
Warburg, 1956 |
Metabolic |
Increased glycolysis and defective mitochondria |
2 |
|
Stevens, 1964 |
Stem-ness |
Origin of cancer stem cells |
11 |
|
Knudson, 1971 |
Genetic |
Tumor suppressor genes: 2-hit hypothesis |
6 |
|
Vogelstein, 1988 |
Genetic |
Multistep carcinogenesis |
4 |
|
Pierce, 1994 |
Stem-ness |
Maturation arrest of stem cells |
12,13 |
|
Seyfried, 2010 |
Metabolic |
Nutrition and cancer |
3 |
|
|
|
|
|
Rous P. A sarcoma of the fowl transmissible by an agent separable from the tumor cells. J Exp Med 1911;13:397-411.
Farber, S. & Diamond, L. K. Temporary remissions in acute leukemia in children produced by folic acid antagonist, 4-aminopteroyl-glutamic acid. N. Engl. J. Med. 238, 787–793 (1948).
Stevens LC. Experimental production of testicular teratomas in mice. Proc Natl Acad Sci U S A. 1964;52:654-661.
Knudson AG Jr. Mutation and cancer: statistical study of retinoblastoma. Proc Natl Acad Sci USA 1971; 68:820-3
Pierce GB, Dixon FJ Jr. Testicular teratomas I. Demonstration of teratogenesis by metamorphosis of multipotential cells. Cancer. 1959;12:573-583.
Sell S, Pierce GB. Maturation arrest of stem cell differentiation is a common pathway for the cellular origin of teratocarcinomas and epithelial cancers. Lab Invest. 1994;70:6-22.
- The subparagraphs discussed in the article are described, in my opinion, in a very general way and give the impression of being unsystematic. We need a general scheme by which the material can be tracked.
We have outlined a general scheme in Introduction so that Unified Theory of Cancer (Section 3.0), Cellular and Cancer Metabolism (Section 4.0), Metabolic Substrates (Section 5.0), Clinical Implications (Sections 6.0 and 7.0), and Drug vs Therapy Development (Section 8.0) can be tracked in the subsequent text, as recommended.
Page 3, Introduction, paragraph 3: “In this Perspective, we re-examine cancer metabolism according to a unified theory and a stem cell origin of cancer. We reassess different types of cellular and cancer metabolism, e.g., glycolysis and oxidative phosphorylation (OXYPHOS) and their metabolic substrates, e.g., glucose, glutamine, arginine, in various cell types, e.g., progenitor CSCs and progeny non-CSCs (Figure 1). We revisit the clinical implications of cellular and cancer metabolism in an obesity epidemic, the aspartame debate, and a ketogenic diet. We illustrate that a pertinent scientific theory about the origin of cancer and of cancer metabolism influences the direction of cancer research as well as the design of drug versus therapy development in cancer care.”
- The drugs given are given without references to clinical trials.
Although details about the clinical trials are provided in the references listed below, we have explicitly mentioned them in the text to differentiate experimental results from clinical evidence, as recommended.
Clinical trials
CAPS2 [58]
STEP 1 [64]
SURMOUNT-1 [65]
SWOG S0003 [101]
TROG 02.02, HeadSTART [102]
LITESPARK-005 [104]
- We need to work on the style of presentation, for example, remove repeated words, for example, in section 3, three paragraphs begin the same way.
Thank you for your help to improve the style of our presentation! Specifically, on page 5, Section 3.1:
“Because stem cells are primeval and tend to be quiescent cells,”
“Considering that (replaced Because) embryonic and germ cells are prototype stem cells,”
“Given that (replaced Because) progenitor stem cells and progeny differentiated cells in both normal tissues and malignant tumors…”
We experimented this Perspective article with a touch of the narrative “poetic” style rather than the conventional “prosaic” style. Hopefully, it appeals to a broader readership that includes scientists and non-scientists (e.g., clinicians, patients, laypeople, etc.).
Small notes:
- What does "5.0 Mitochondrial Switch" mean on line 156?
Section 4.2 Mitochondrial Switch introduces the idea of mitochondrial or metabolic switch as published by Kim et al “a hypoxia-induced metabolic switch that shunts glucose metabolites from the mitochondria to glycolysis.”
Added clarification with reference in section 4.3, line 189: “HIF1α is a master regulator and metabolic switch [Kim, 2006]…”
Kim JW, Tchernyshyov I, Semenza GL, Dang CV. HIF-1-mediated expression of pyruvate dehydrogenase kinase: a metabolic switch required for cellular adaptation to hypoxia. Cell Metab 2006; 3:177-85.
- I’m not sure that an epigraph is appropriate.
The epigraph is meant to connect with the prologue, “History is just new people making old mistakes (Sigmund Freud).” and summarize how we may learn from history about cancer metabolism and in science (section 2.0 and Table 1).
We modified the epigraph to make sure that it is appropriate for this Perspective article: “When we formulate the right scientific theory and adopt or adhere to the proper scientific method in scientific research, it will be less likely if not unlikely that we will continue to make the same old mistakes or commit more new mistakes.”

Reviewer 2 Report (Previous Reviewer 2)
Comments and Suggestions for Authors
As I stated in the first round of review (cancers-2781741), I believe the manuscript offers valuable insights into the complex interplay between cancer and cellular metabolism, highlighting the potential for novel therapeutic approaches based on the stem cell theory of cancer. The revision makes it more solid and suitable for a perceptive article.
Author Response
We thank the reviewers for your insightful comments and suggestions. We have provided our responses below in bold with changes highlighted in red underneath the reviewer comments. We have incorporated all changes recommended by the reviewers with the changes highlighted in red in the manuscript.
As I stated in the first round of review (cancers-2781741), I believe the manuscript offers valuable insights into the complex interplay between cancer and cellular metabolism, highlighting the potential for novel therapeutic approaches based on the stem cell theory of cancer. The revision makes it more solid and suitable for a perceptive article.
We thank the reviewer for your expert insights and favorable comments!
Reviewer 3 Report (New Reviewer)
Comments and Suggestions for Authors
The manuscript “Stem Cell Theory of Cancer: Clinical Implications for Cellular Metabolism and Anti-Cancer Metabolomics” is a perspective article regarding the different types of cellular metabolism in CSC and non-CSC and possible implications.
The paper is not considered good in terms of content and composition, and also shows crucial flaws. For all these reasons, my recommendation is to reject.
1. This manuscript seems a mix between a review and a perspective; it is excessively long for being a pure perspective.
2. The manuscript is so confused that the ideas of authors disappear. It is not clear what they want to say and the final utility of this manuscript. The flow of the thoughts in the article is not smooth.
3. The construction of sentences is excessively and uselessly complex, intricated and this result in difficulties for the reader to follow what the authors want to say. I would honestly suggest a rewriting of the whole manuscript in a clearer way, avoiding very long and complex periods. On the contrary, sometimes (e.g. abstract) the sentences used are so simplistic that could not be used in a scientific paper.
4. In addition, several important aspects are described in a very superficial and non-informative manner.
5. The paper fails to present innovative ideas compared to existing review papers on similar topics.
6. Large parts of the manuscript are in red, suggesting a previous rejection?!
7. There are gross errors, such as in figure 1 “Kreb Cycle” instead of Krebs Cycle. It is reported a 4C fatty acid while the most common in humans are ranging between 12C and 24C.
8. The manuscript completely ignores a master regulator of metabolism and nicotinamide homeostasis, namely nicotinamide N-methyltransferase. The activity of this enzyme affects the metabolism of cancer cells, including CSC where it was found to be overexpressed. Many NNMT inhibitors have been developed which could be used for targeting cancer cell metabolism and CSC metabolism (PMID: 34572571; PMID: 34704059; PMID: 34424711).
Comments on the Quality of English LanguageExtensive editing of English language required
Author Response
Reviewer 3
The manuscript “Stem Cell Theory of Cancer: Clinical Implications for Cellular Metabolism and Anti-Cancer Metabolomics” is a perspective article regarding the different types of cellular metabolism in CSC and non-CSC and possible implications.
The paper is not considered good in terms of content and composition, and also shows crucial flaws. For all these reasons, my recommendation is to reject.
- This manuscript seems a mix between a review and a perspective; it is excessively long for being a pure perspective.
We understand that all reviewers (including Reviewer 3) expect some details and comprehensiveness, which we try our best to provide in this Perspective, and which contribute to the length of this article. However, we managed to make sure that it does meet all requirements specified by Cancers, including 3,000 words at a minimum and more than 30 references.
Furthermore, we have condensed sections 7.1.1.1 to 7.1.1.6 to a single phrase in section 7.1.1, lines 464-5 for the sake of brevity and unity, as recommended:
But is anti-metabolism the only way metformin tackles cancer, or are its anti-cancer effects operative through multiple ways and pathways (e.g., anti-metabolic [90,91], anti-CSC [92,93], ant-EMT [94,95], pro-senescent [96], pro-differentiating [97], and/or multi-pronged [98-102]?
- The manuscript is so confused that the ideas of authors disappear. It is not clear what they want to say and the final utility of this manuscript. The flow of the thoughts in the article is not smooth.
We agree that a broad subject with an expansive variety of topics concerning cellular and cancer metabolism can be overwhelming to some readers. However, we have reorganized and restructured this paper, as recommended by the other reviewers, who are either never confused or no longer confused.
Hopefully, a condensed version involving sections 7.1.1.1 to 7.1.1.6 further improves the flow and fluency of this paper, as recommended.
- The construction of sentences is excessively and uselessly complex, intricated and this result in difficulties for the reader to follow what the authors want to say. I would honestly suggest a rewriting of the whole manuscript in a clearer way, avoiding very long and complex periods. On the contrary, sometimes (e.g. abstract) the sentences used are so simplistic that could not be used in a scientific paper.
We experimented this Perspective article with a touch of the narrative “poetic” style rather than the conventional “prosaic” style. Hopefully, this is an effective way to deliver a subliminal rather than literal message. Sometimes, when the idea may be premature and the data is preliminary, perhaps we need a little prose than logic to link ideas and integrate data. If it concerns an important concept, we hope that the message rather than the style will appeal to a broader readership that include both scientists and non-scientists, clinicians, patients, and their caregivers.
- In addition, several important aspects are described in a very superficial and non-informative manner.
We appreciate the reviewer’s previous comments regarding brevity and clarity in a Perspective article. Hopefully, the pertinent references will provide some of the detailed information some readers may be looking for without making this article even longer than it already is.
- The paper fails to present innovative ideas compared to existing review papers on similar topics.
Currently, nobody has yet applied a unified theory of cancer to elucidate cancer metabolism that we are aware of. To reexamine cancer metabolism under a singular premise of a unified theory of cancer is innovative. Many scientific treatises show preclinical data but lack clinical evidence and do not distinguish a crucial difference between Drug versus Therapy Development (section 8.0), which also makes this Perspective novel and unique in our opinion.
In addition, one of the main purposes of this Perspective is to update the readers about recent publications and observations pertaining to cancer metabolism. For example, addressing the ongoing controversy regarding aspartame (by Gnudi, Ann Glob Health 2023 [59]), ketogenic diet (by Chi, Prostate Cancer Prostatic Dis 2022 [67]), and GLP-1 agonists (by Sforza, Cardiovasc Diabetol 2022 [79]) in the causation or prevention of cancer with respect to cancer metabolism is unprecedent, and could be groundbreaking, from our perspective.
- Large parts of the manuscript are in red, suggesting a previous rejection?!
Fortunately, the previous reviewers have provided some objective and constructive recommendations that we have adopted in our current version, and they have favorably accepted our responses to their comments.
- 7. There are gross errors, such as in figure 1 “Kreb Cycle” instead of Krebs Cycle. It is reported a 4C fatty acid while the most common in humans are ranging between 12C and 24C.
We thank the reviewer for pointing out a typo in “Krebs” Cycle, which we have corrected in Figure 1.
We have also modified the figure to illustrate the most common saturated fatty acid found in the human body, palmitic acid (16C).
- The manuscript completely ignores a master regulator of metabolism and nicotinamide homeostasis, namely nicotinamide N-methyltransferase. The activity of this enzyme affects the metabolism of cancer cells, including CSC where it was found to be overexpressed. Many NNMT inhibitors have been developed which could be used for targeting cancer cell metabolism and CSC metabolism (PMID: 34572571; PMID: 34704059; PMID: 34424711).
We thank the reviewer for suggesting inclusion of another master regulator of metabolism and nicotinamide homeostasis that involves NNMT in the section of Metabolic Substrates (section 5.4) in our discourse about cancer metabolism to ensure comprehensiveness and balance.
5.4. S-adenosyl-L-methionine (SAM)
In early development, consumption of SAM by nicotinamide N-methyltransferase (NNMT) makes it unavailable for histone methylation, resulting in an altered epigenetic landscape within hESC [Sperber, 2015. In some cancers, NNMT mediates EMT [Cui, 2019] and induces resistance to apoptosis via the mitochondrial pathway [Zhang, 2014].
In addition, the activity of NNMT is tightly linked to the maintenance of nicotinamide adenine dinucleotide (NAD+), which modulates multiple enzymatic reactions that affect redox metabolism, mitochondrial functions, stemness properties, autophagic processes, cellular stress, ion homeostasis, and the circadian rhythm [Covarrubias, 2021, Katsyuba, 2020]. This is reminiscent of a complex metabolic network in need of a unified conceptual framework, as premised in this Perspective [14,15]. Importantly, NNMT may be a potential biomarker and therapeutic target for cancer diagnosis and treatment [Roberti, 2021; Wang, 2022; Van Haren, 2021].
Sperber H, Mathieu J, Wang Y, Ferreccio A, Hesson J, Xu Z, Fischer KA, Devi A, Detraux D, Gu H, et al. The metabolome regulates the epigenetic landscape during naive-to-primed human embryonic stem cell transition. Nat. Cell Biol 2015; 17:1523-1535.
Zhang J, Wang Y, Li G, Yu H, Xie X. Down-regulation of nicotinamide N-methyltransferase induces apoptosis in human breast cancer cells via the mitochondria-mediated pathway. PLoS One 2014; 9:e89202.
Cui Y, Zhang L, Wang W, Ma S, Liu H, Zang X, Zhang Y, Guan F. Downregulation of nicotinamide N-methyltransferase inhibits migration and epithelial-mesenchymal transition of esophageal squamous cell carcinoma via Wnt/beta-catenin pathway. Mol Cell Biochem 2019; 460:93-103.
Covarrubias AJ, Perrone R, Grozio A, Verdin E. NAD+ metabolism and its roles in cellular processes during ageing. Nat Rev Mol Cell Biol 2021; 22:119-141
Katsyuba E, Romani M, Hofer D, Auwerx J. NAD+ homeostasis in health and disease. Nat Metab 2020; 2:9-31
Roberti A, Fernandez AF, Fraga MF. Nicotinamide N-methyltransferase: At the crossroads between cellular metabolism and epigenetic regulation. Molecular Metabolism 2021; 45:101165.
Wang W, Yang C, Wang T, Deng H. Complex roles of nicotinamide N-methyltransferase in cancer progression. Cell Death & Disease 2022; 13:267.
Van Haren MJ, Gao Y, Buijs N, Campagna R, Sartini D, Emanuelli M, Mateuszuk L, Kij A, Chlopicki S, Martinez de Castilla PE, et al. Esterase-sensitive prodrugs of a potent bisubstrate inhibitor of nicotinamide N-methyltransferase (NNMT) display cellular activity. Biomolecules 2021; 11:1357.
Please review the level of English and decide whether revisions are necessary. We propose that you have your manuscript reviewed by an experienced English-speaking colleague. Fortunately, we have not had issues with the level of English with the other reviewers and in our previous publications by Cancers. Nevertheless, we have asked one of our co-authors, Dr. Timothy Langford, Professor and Chairman of the Department of Urology, who is a native English speaker, to review and approve the quality of English writing in this manuscript, as proposed.

Round 2
Reviewer 3 Report (New Reviewer)
Comments and Suggestions for Authors
The reject recommendation cannot be changed also upon manuscript revision.
Comments on the Quality of English LanguageExtensive editing of English language required.
This manuscript is a resubmission of an earlier submission. The following is a list of the peer review reports and author responses from that submission.
Round 1
Reviewer 1 Report
Comments and Suggestions for Authors
This topic offers the potential for an interesting overview, but some sections read more like a general discussion than a scientifically rigorous text. The content appears to be part of a series of papers (doi: 10.3390/cancers15235533; doi: 10.3390/cancers15225385; doi: 10.3390/cancers15092516), suggesting a repetitive presentation of data centered around a single concept. There are also redundancies with content available at https://www.mdpi.com/2073-4409/12/23/2686.
The only figure in the text is not original but was copied with permission from another source. In addition, the only table is scarce and not very informative. As an overall, it adds only minimally to the advancement of knowledge in the area and merely adds redundant information to the scientific literature.
Reviewer 2 Report
Comments and Suggestions for Authors
The manuscript presents a perspective on the stem cell theory of cancer and its implications for understanding and targeting cancer metabolism. It emphasizes the impact of critical metabolic pathways on cancer progression, such as glycolysis, mitochondrial activity, and hypoxia. Additionally, it discusses the potential therapeutic implications of targeting cancer metabolism. Overall, the manuscript offers valuable insights into the complex interplay between cancer and cellular metabolism, highlighting the potential for novel therapeutic approaches based on the stem cell theory of cancer. More explicit guidance on practical applications and potential future research directions would enhance the manuscript’s value to the broader scientific community.
Reviewer 3 Report
Comments and Suggestions for Authors
There is currently no unified theory of the origin of cancer that clarifies whether cancer is a metabolic, genetic, or stem cell disease. The authors consider different types of cellular metabolism, such as glycolytic and mitochondrial, with glucose, glutamine, arginine and fatty acids in RSCs and non-ROCs. The authors suggested that the relevant scientific theory of the origin of cancer and cancer metabolism influences the direction of cancer research, as well as the development of drugs and cancer treatments.
1. The article does not have an abstract. Add please.
2. There are not enough pivot tables and/or diagrams. So, I would at least add others to the historical excursion.
3. The subparagraphs discussed in the article are described, in my opinion, in a very general way and give the impression of being unsystematic. We need a general scheme by which the material can be tracked.
4. The drugs given are given without references to clinical trials.
5. We need to work on the style of presentation, for example, remove repeated words, for example, in section 3, three paragraphs begin the same way.